# Prevalence of Postoperative Atrial Fibrillation and Impact to Nursing Practice—A Cross Sectional Study

**DOI:** 10.3390/medsci11010022

**Published:** 2023-03-03

**Authors:** Ana Brčina, Kristian Civka, Renata Habeković, Sabina Krupa, Ana Ljubas, Wioletta Mędrzycka-Dąbrowska, Adriano Friganović

**Affiliations:** 1Department of Anethesiology and Intensive Medicine, University Hospital Centre Zagreb, Kišpatićeva 12, 1000 Zagreb, Croatia; 2Department of Nursing, University of Applied Health Sciences Zagreb, Mlinarska Cesta 38, 10000 Zagreb, Croatia; 3Institute of Health Sciences, College of Medical Sciences of the University of Rzeszow, St. Warzywna 1A, 35-310 Rzeszow, Poland; 4Department of Anaesthesiology Nursing & Intensive Care, Faculty of Health Sciences, Medical University of Gdansk, Dębinki 7, 80-211 Gdańsk, Poland; 5Faculty of Health Studies, Department of Nursing, University of Rijeka, Ul. Viktora cara Emina 5, 51000 Rijeka, Croatia

**Keywords:** atrial fibrillation, cardiac surgery, nurses, prevalence, postoperative

## Abstract

Background: Atrial fibrillation is the most common clinically significant cardiac arrhythmia, and it might lead to heart failure, which prolongs the duration of hospitalization and consequently increases the cost of treatment. Thus, diagnosing and treating atrial fibrillation should be the first line of defense against further complications. This study aimed to determine the incidence rate of postoperative atrial fibrillation and correlation with cardiac surgery on heart valves. A specific aim was to determine the relationship between the prevalence of atrial fibrillation and socio-demographic features. Methods: The study has a prospective cross-sectional design. The questionnaire was anonymous, requesting socio-demographic information as inclusion criteria, and the data were analyzed using descriptive statistics methods. Results: The sample was 201 patients. χ^2^ test and *t*-test were performed where we found that the frequency of atrial fibrillation was higher in the groups that have had valve surgery compared to other cardiac surgeries (χ^2^ = 7.695, ss = 2, *p* = 0.021). Atrial fibrillation increased with the age of the patients, but the prevalence of atrial fibrillation was not correlated with body weight. Conclusion: The results of this this study show that atrial fibrillation was higher in the participants who had valve surgery compared to other cardiac surgeries. There was also an increase in atrial fibrillation in the older participants. The results of this study can help to improve nursing practice and the quality of care for cardiac surgery patients with regard to daily activities, or planning nursing care due to the patient’s condition.

## 1. Introduction

Atrial fibrillation (AF) is the most common clinically significant cardiac arrhythmia. It is thought that, by 2060, 17.9 million people in Europe will suffer from AF [1]. It is a major cause of morbidity, mortality, and prolonged healthcare in cardiac surgery patients. The onset of AF is influenced by numerous factors such as age, race, and gender; the prevalence of AF generally increases with age. Conditions such as heart valve disease, diabetes mellitus, arterial hypertension, obesity [2], sleep apnea, and inflammation [3] are risk factors for the occurrence of AF. Although uncommon, rheumatic arterial heart disease is significantly associated with the development of AF [4]. It is assumed that the prevalence of AF after cardiac surgery ranges from 15% to 40% in coronary revascularization surgical procedures; from 37% to 60% in valvular surgery interventions; and more than 60% in combined interventions [5]. Of all cases of AF after surgery, 90% of them occur within the first four days of the postoperative period [6,7].

Once a diagnosis of atrial fibrillation is confirmed, nurses together with doctors have an important role in explaining the condition to the patients in a way they can understand [8,9]. When assessing the patient’s condition, nurses collect data on the existence of palpitations, chest pain, shortness of breath, and dizziness [10]. Nurses had to assess other risk factors for developing AF such as excessive alcohol consumption, obesity, smoking, drug abuse, hyperlipidemia, previous rheumatic heart disease, and a history of valvular heart disease or ischemic heart disease [10,11,12,13,14,15]. 

Furthermore, identifying patients at high risk of developing of AF may allow for the modification of risk factors to reduce the postoperative atrial fibrillation (POAF) burden [16,17,18]. Nurses, as independent professionals, work together with other members of the healthcare team to meet a wide range of patient needs [19]. Since nurses are the first to be in contact with the patient, they must know the pathophysiology and symptoms of AF [20]. The early recognition of symptoms and a timely response to the occurrence of AF significantly reduces the possible complications of AF and increases patients’ qualities of life [21,22,23,24].

The main goals in the treatment of AF are to maintain adequate cardiac volume and tissue perfusion, and to prevent thromboembolism. The recognition of irregular rhythm and P-wave deficiency initiates the treatment of AF. Appropriate interventions such as measuring vital signs, properly positioning the patient in a semi-recumbent position, and the administration of oxygen can significantly reduce the risk of heart failure. The appropriate use of oral anticoagulation is an established pillar of AF management to reduce stroke risk [22]. With proper information about the occurrence of AF in the postoperative period, nurses can obtain valuable information and prepare specific plans for nursing care. An individual approach for specific group of patients may improve nursing care. 

The aim of this study was to show the association of the prevalence of post-operative atrial fibrillation with variables of gender, type of surgery, age, and body weight. A specific aim was to determine the relationship between the prevalence of atrial fibrillation and socio-demographic features. There are many other studies involving POAF in cardiac surgery, but there is no adequate research in Croatia. We believe that our study’s findings will produce a scientific contribution to nursing research and improve nursing practice and the quality of nursing care for patients [23]. This research will bring new knowledge to evidence-based nursing in Croatia, and for nursing in general [24].

## 2. Materials and Methods

### 2.1. Study Design

The research study “Prevalence of Postoperative Atrial Fibrillation and Impact to Nurses Practice—A Cross Sectional Study” was a prospective cross-sectional study conducted on all consecutive patients of the Department of Cardiac Surgery in the University Hospital Centre Zagreb May–September 2021.

### 2.2. Ethics and Approval

Prior to conducting the study, on 12 May 2021, the principal investigator sought the approval of the Ethics Committee of the University Hospital Centre Zagreb (Class: 8.1-21/112-2; Number: 02/21JG). The ethical principles and the principles of the Declaration of Helsinki were respected during the study [24]. The personal data, names, and surnames of the respondents were not used in the research. The data collected from the respondents were used only for the purposes of this research, and were stored in such a way that they were available only to the co-authors of this research.

### 2.3. Participants

All the cardiac surgery patients in the observed period of five months were included in the study. The inclusion criteria were signed, informed consent by patients who are 18 years and older; inability or refusal to provide informed consent meant that the patient was excluded. The data on the occurrence of atrial fibrillation was collected for the analysis and confirmation of the correlation in all patients who underwent cardiac surgery at the Department for Cardiac Surgery in University Hospital Centre Zagreb.

### 2.4. Sociodemographic Features

The study instrument was a data form in which the principal investigator collected information about the patients from a medical chart and nursing documentation. The sociodemographic features and clinical data were collected by the investigators of this study.

The follow-up form included demographic data such as age, gender, and education level. Health-related information included the patient’s weight, height, and type of operation; the existence of pre- and postoperative AF; in the case of existing POAF, whether it had been treated and in what way, and whether it had been converted; and, finally, the existence of other diseases.

### 2.5. Statistics

The collected data were entered into a Microsoft Excel file and analyzed using descriptive statistical methods (χ^2^ test and *t*-test); the significance level was *p* < 0.05. For nominal (categorized) variables, the number and percentage of respondents in the corresponding categories were shown, and the statistical significance of the differences was calculated using the χ^2^ test. The SPSS program IBM SPSS Statistics for Windows, Version 25.0 was used for data analysis [25]. 

## 3. Results

### 3.1. Sociodemographic Features

A total of 201 participants were included in this study. The data about the patients were collected from medical chart and nursing documentation. The socio-demographic data were age, gender, and education level, and the health data included body weight, height, and type of operation. The mean age of the participants was 75 ± 5 (35.3%) years, and 69.8% of the participants were men. Table 1 describes the baseline characteristics. 

Table 1 represents the sociodemographic features of the participants included in this study and the type of cardiac surgery they underwent. 

### 3.2. Patient Data

Table 2 presents the data on the prevalence of preoperative and postoperative atrial fibrillation. 

Table 2 represents the data regarding the prevalence of AF in patients included in this study and the treatment in the case of its existence.

The research included data on the existence of diseases such as arterial hypertension in 153 patients (76.2%), coronary heart disease in 95 patients (47.5%), and rheumatic heart disease in 3 patients (1.3%), and all the other conditions that were present in the participants; 97% (195) of the participants had other diseases. Table 3 the represents prevalence of other diseases.

Table 4 represents the data regarding the prevalence of postoperative AF in patients by gender.

The χ^2^ test did not find any statistically significant differences between male and female patients in the prevalence of postoperative AF (χ^2^ = 0.405, ss = 1, Fisher’s exact *p* = 0.541). 

Table 5 represents the data regarding the prevalence of postoperative AF in patients by the type of cardiac surgery.

The χ^2^ test showed that the prevalence of postoperative AF differs according to the type of surgery; the frequency of AF was higher in the groups that had had valve surgery compared to other cardiac surgeries (χ^2^ = 7.695, ss = 2, *p* = 0.021). 

Figure 1 represents the prevalence of POAF by age, whereas most participants between 70 and 79 years of age have had POAF, and most participants aged from 80 to 89 years of age did not develop POAF. 

The point—biserial correlation coefficient showed that the association of age with the prevalence of AF is low (r = 0.22, *p* < 0.001), which means that there is a greater likelihood of AF in older age groups (an increase in AF with the age of patients).

Figure 2 represents prevalence of postoperative AF by body weight, whereas most participants between 81 and 90 kg had AF. They are followed by 26 participants who weighed between 71 and 80 kg but did not develop AF. 

The point—biserial correlation coefficient did not show the correlation of body weight with the AF prevalence (r = −0.08, *p* = 0.292).

## 4. Discussion

This study aimed to determine the incidence rate of postoperative atrial fibrillation and its correlation with cardiac surgery on heart valves in nursing practice. We also aimed to show the association of the prevalence of postoperative atrial fibrillation with variables of gender, age, and body weight. The results of this study showed that the prevalence of postoperative AF differs according to the type of surgery; the frequency of AF was higher in groups that previously had valve surgery compared to other cardiac surgeries (χ^2^ = 7.695, ss = 2, *p* = 0.021). It is also confirmed that there is an increase in AF with the age of patients. However, there is no correlation of body weight with the AF prevalence.

The prevention of AF should begin at the earliest possible age, considering that the prevalence of AF is known to increase with advancing age. The prevention of AF should be crucial at the primary and secondary levels in cardiovascular disease, but primordial and primary prevention are fundamental in young adults [26].

A study conducted in France showed that AF is more common in women and has a greater risk factor for cardiovascular diseases and death in women than in men [27]. However, the risk of developing cardiovascular disease increases with age in both genders. The increased risk of AF with old age appears to be greater in men, although not all studies have presented sex differences. However, although older men may be at a greater risk of developing AF, it is associated with worse outcomes in women, including stroke [28,29]. Our research included 69.8% men and 30.2% women (Table 1).

Gleason et al. conducted a study examining the association of sex, age, and education level with patient-reported outcomes (AF-related quality of life, symptom severity, and emotional and functional status). The results of this study showed that women, younger adults, and individuals with lower levels of education reported comparatively poorer outcomes [30]. On the other hand, our results in Table 1 showed that many of the respondents in our study had high school education (62.0%), while the smallest number of respondents had university education (7.5%). 

A study conducted by Andersen et al. showed that body size is a very important indicator of the risk of developing AF. Greater height and weight are strongly associated with a higher risk of atrial fibrillation. The mechanisms remain unknown but may involve an increased atrial volume load with larger body size [31]. The average weight of our respondents was 81–90 kg (27.7%), while the average height was 171–180 cm (34.2%), which can be seen in Table 1. We did not find a correlation between the body weight and AF prevalence.

According to the research of Baeza-Herreras et al., the prevalence of AF after cardiac surgery ranges from 15 to 40% in coronary revascularization surgical procedures and from 37 to 60% in valvular surgery interventions, and it is higher than 60% in the combined interventions [6]. It occurs in 24% of the patients undergoing a heart transplant. In this study, we particularly focused on the occurrence of AF after valve surgery, which was received by 69 subjects (34.2%) (Table 1).

A literature search found that the prevalence of POAF is 20–40% [32], but, in our study, as many as 46.8% of the respondents developed AF in the postoperative period, as is shown in Table 2. 

Recent research suggests that the decision to treat AF is made by the physician in consultation with the patient, and it is primarily based on managing the patient’s symptoms and preferences. It has been reported in the literature that there are gender differences in these management strategies, i.e., which treatment is recommended and the response to therapy [33,34,35,36]. However, in 94.7% of our subjects AF was treated when it occurred after surgery. In 5.3% of the subjects, it was not treated because they had permanent AF that did not respond to drug treatment even before the surgery. In Table 2, we can see that most AF arising in the early postoperative period was converted to a sinus rhythm (89.5%). This was most often achieved with the use of drugs (83.2%).

The prevalence of arterial hypertension can be up to 80% in individuals older than 65 years, and 26% in adults younger than 45 years old. Hypertension leads to cardiovascular complications, including coronary heart disease, and heart failure that consequently leads to atrial fibrillation and mortality [26,37,38,39]. 

Lee et al. investigated the relationship between the arterial hypertension burden and the development of incident AF. The results of their study showed that subjects with arterial hypertension burdens were associated with an increased risk of 8–27% for incident AF [40]. In our study, 76.2% of the subjects had arterial hypertension.

A strong correlation between AF and coronary heart disease has been reported [41]. In a study conducted by Ferreira et al., 241 patients (4.8%) developed AF during the follow-up. Older age, LVEF > 35%, a history of PCI and CABG, white race, a SBP < 110, and a higher BMI were independently associated with the risk of new onset AF [42]. Coronary heart disease was present in 47.5% of the participants in our study.

In recent decades, there has been increasing interest in the association of AF with rheumatic heart disease. Recent studies have found that there is an association between rheumatic heart disease and high rates of disability and premature death across African and Asian countries [43]. Older age and the presence of mitral valve disease (special stenosis) are significantly associated with the development of atrial fibrillation [42]; 40–75% of individuals with mitral stenosis have AF [12]. Rheumatic heart disease is not a common disease, and only 1.3% of our participants had it.

Almost all our subjects, 97% of them, had other diseases. Hyperlipidemia was as common as coronary heart disease, being present in 48% of our subjects. Hyperlipidemia is in itself a risk factor for the development of heart disease. The next most common disease was diabetes mellitus, which was present in 32% of our subjects. The pathophysiology of diabetic-related AF is not fully understood, but it is related to structural, electrical, electromechanical, and autonomic remodeling [44]. AF is very common in patients with severe aortic stenosis [45]; aortic valve stenosis was present in 20% of our subjects.

We aimed to show the difference in the incidence of AF in women and men. The results showed that there was no statistically significant difference in the incidence of POAF in women and men (Table 4). Numerous previous studies have indicated that there are significant gender differences in the epidemiology, pathophysiology, and therapeutic outcome of AF, which provided us with strong reasons for further research. A literature review showed that, in general, AF is more prevalent in men than in women in age-matched observations [46,47,48].

Our next goal was to determine which surgeries most commonly caused AF; the results listed in Table 5 showed that the incidence of AF was higher in the group of subjects who had heart valve surgery than in the groups who had other cardiac surgeries or a combination of heart valve surgery and other cardiac surgeries. A study entitled “Evaluation of the incidence of new atrial fibrillation after aortic valve replacement”, conducted over a period of three years, showed that fibrillation occurs in about 50% of the patients hospitalized for transcatheter aortic valve implantation and aortic valve replacement. Hospitalizations due to newly developed atrial fibrillation are associated with increased in-hospital mortality [14]. Another study demonstrated that there are predictors that indicate whether new-onset AF is likely to occur after surgery. In a study involving 2261 subjects who underwent mitral valve surgery over a 10-year period, the prevalence of AF occurring more than 90 days after surgery in patients who did not have AF prior to surgery was examined. AF was found to occur in 14% of the subjects after 5 years and in 23% of subjects after 10 years. The patients who had degenerative mitral regurgitation were less likely to develop AF. Multivariable factors for the development of AF are tricuspid valve surgery, aortic valve surgery, and older age. New-onset AF did not affect overall survival [49].

Another aim of our research was to show the association of POAF with patient age. The results shown in Figure 1 show that the prevalence of POAF increases with advancing age. In AF, many factors work over the years. For example, a chronic subclinical inflammatory response, defined as continuous weak activation of the systemic immune response, is characterized by the biological aging of organ systems. Both age and AF are associated with elevated concentrations of reactive oxygen species. Furthermore, inflammation is associated with endothelial dysfunction and collagen catabolism, resulting in an increase in TGF-ß1 (transforming growth factor) activity [11].

Despite the fact that numerous recent studies have shown that there is a strong link between the development of AF and obesity, our results did not show that there was correlation between body weight and AF prevalence (Figure 2). The research conducted by the Framingham Heart Study states that each unit of increase in body mass index (BMI) is associated with an increase in the risk of AF by 4–5%, independent of other comorbidities such as acute myocardial infection, diabetes, and other conditions [50].

Despite the significant impact of atrial fibrillation on health care and on the population, it has been generally considered a benign disorder in clinical practice and is often not seen as being as important as ventricular arrhythmias in clinical care nor in health care research [51]. By conducting this study, we wanted to emphasize the importance of the incidence of AF in patients and, accordingly, the impact on nursing practice during AF management. Nurses are integral to the care of patients with AF. It is essential for nurses to stay informed of current guidelines and new evidence so that the assessment, management, and education of the AF patients and their families can be optimized [52]. This study impacted nurses’ knowledge of atrial fibrillation and anticoagulation, and influenced their uptake and use of AF risk scales assessment tools in clinical practice. Future research should focus on whether a similar intervention might improve patient-centered outcomes such as patients’ knowledge of their condition and therapies, medication adherence, time in the therapeutic range, and quality of life.

Many different studies suggested that nursing staffing practice in the postoperative period (adequate staffing level, adequate skills, and educational level) are connected with lower rates of surgical mortality and lower numbers of adverse events [53]. With the better assessment of potential complications, we can assure there is adequate nurses staffing to better monitor patients [53]. The estimation of staffing practice with POAF occurrence is important to help hospital managers determine the number and educational degree of nurses needed for the patient after cardiac surgery procedure with the aim to reduce postoperative complications [53,54]. Postoperative cardiac events are frequent complications of surgery, and their occurrence could be associated with suboptimal nurse staffing practices, but the existing evidence remains scattered [55]. Higher nurse staffing levels, higher registered nurse education (baccalaureate degree level), and more supportive work environments provide better patient safety [55]. The existing evidence regarding postoperative cardiac events is limited, which warrants further investigation. These facts also provide scientific potential for future research associated with nursing degrees and adverse events.

## 5. Study Limitations

The biggest obstacle to this study was the ongoing SARS-CoV-2 pandemic that limited the number of hospitalized patients and their participation in this study. Additionally, this resulted in a reduced number of cardiac operations. Furthermore, there are not enough studies conducted in Croatia and worldwide regarding AF during the postoperative period of cardiac patients. Our study did not include children or infants. On the other hand, the patient data in this study are from 2021 and the sample size is small, suggesting that repeating the study with larger sample would be productive.

## 6. Conclusions

This research paper summarizes the development of AF in the postoperative period of patients who had valve operations at the Department for Cardiac Surgery in the University Hospital Centre Zagreb. Our results showed that there was no correlation between body weight and AF prevalence but found that the frequency of AF was higher in the groups that had valve surgery compared to other cardiac surgeries. The results of this study can help to improve nursing practice and the quality of care for cardiac surgery patients with regard to daily activities for patients or planning nursing care due to the patient’s condition. The data regarding POFA can impact nursing staffing and assure the safety of care.

## Figures and Tables

**Figure 1 medsci-11-00022-f001:**
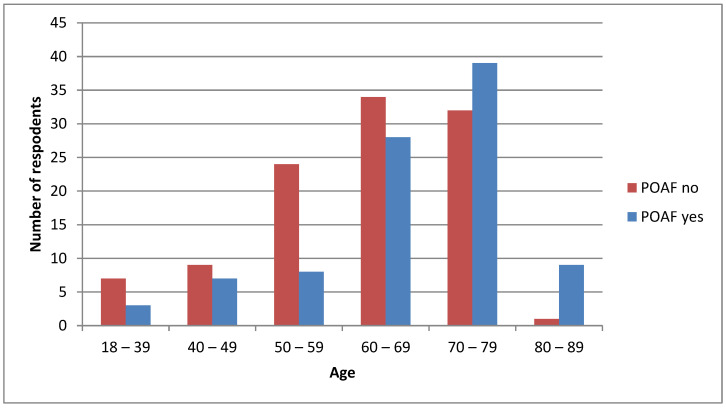
Prevalence of postoperative AF by age.

**Figure 2 medsci-11-00022-f002:**
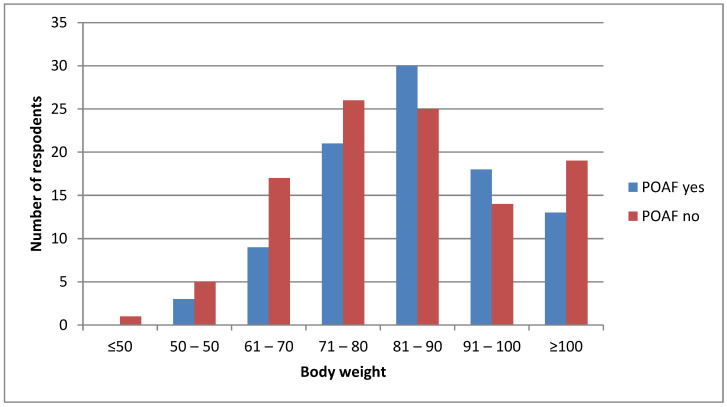
Prevalence of AF by body weight.

**Table 1 medsci-11-00022-t001:** Sociodemographic features of study participants.

		N	(%)
Age	18–39	10	(4.9%)
	40–49	16	(7.9%)
	50–59	32	(15.9%)
	60–69	62	(30.8%)
	70–79	71	(35.3%)
	80–89	10	(4.9%)
Gender	Male	140	(69.7%)
	Female	61	(30.3%)
Education level	Secondary school education	116	(62.0%)
	College education	14	(7.4%)
	University degree	33	(17.5%)
	Primary school	23	(12.7%)
Weight/kg	≤50	1	(.5%)
	51–60	8	(4.0%)
	61–70	26	(12.9%)
	71–80	46	(22.8%)
	81–90	56	(27.8%)
	91–100	32	(15.9%)
	≥100	32	(15.9%)
Height/cm	≤150	5	(2.5%)
	151–160	18	(9.0%)
	161–170	62	(30.8%)
	171–180	69	(34.3%)
	181–190	42	(20.8%)
	191–200	4	(2.0%)
	≥200	1	(.5%)
Type of operation	Valve operation	69	(34.3%)
	Other cardiac surgery	100	(49.7%)
	Valve operation and other cardiac surgery	32	(15.9%)

**Table 2 medsci-11-00022-t002:** Prevalence of atrial fibrillation.

		N	(%)
Preoperative AF	Yes	35	(17.4%)
	No	166	(82.6%)
Postoperative AF	Yes	94	(46.8%)
	No	107	(53.2%)
Was the AF treated?	Yes	90	(94.7%)
	No	5	(5.3%)
Was the AF converted?	Yes	85	(89.5%)
	No	10	(10.5%)
How was the AF converted?	Spontaneously	6	(6.3%)
	Medical treatment	79	(83.2%)
	It did not convert	10	(10.5%)

**Table 3 medsci-11-00022-t003:** Prevalence of other diseases besides arterial hypertension, coronary heart disease, and rheumatic heart disease.

Disease	N	(%)
Hyperlipidemia	94	(46.7%)
Diabetes mellitus	63	(31.3%)
Aortic valve stenosis	40	(19.4%)
Nicotinism	19	(9.4%)
Asthma	10	(4.9%)
Mitral valve insufficiency	9	(4.4%)
Ishemic cardiomyopathy	8	(3.9%)
Endocarditis	6	(2.9%)

**Table 4 medsci-11-00022-t004:** Prevalence of postoperative AF by gender of participants.

		Gender			
		Male		Female	
		N	(%)	N	(%)
Postoperative AF	Yes	68	(48.2)	26	(43.3)
	No	73	(51.8)	34	(56.7)

**Table 5 medsci-11-00022-t005:** Prevalence of postoperative AF by type of operation.

		Operation Type					
		Heart Valve Operation		Other Cardiac Surgery		Heart Valve Operation and Other Cardiac Surgery	
		N	(%)	N	(%)	N	(%)
Postoperative AF	Yes	39	(57.4)	37	(37.0)	18	(54.5)
	No	29	(42.6)	63	(63.0)	15	(45.5)

## Data Availability

Not applicable.

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
