# Peer review of "Prevalence of Postoperative Atrial Fibrillation and Impact to Nursing Practice—A Cross Sectional Study"

_medsci, 2023, doi:10.3390/medsci11010022_

Round 1

Reviewer 1 Report (Previous Reviewer 2)

The manuscript titled " Prevalence of Postoperative Atrial Fibrillation and Impact to Nurses Practice – A Cross Sectional Study " have identified several issues that should be addressed before publication.

 1, The authors emphasize:  This study aimed to determine the incidence rate of postoperative atrial fibrillation and its correlation with cardiac surgery on heart valves in nurses practice. ”, as well as ” this study impacted nurses’ knowledge of atrial fibrillation and anticoagulation”. However, the manuscript does not provide any discussion on how the study results specifically affect nursing practice.

 2, The discussion of the manuscript includes a substantial amount of information on the relationship between atrial fibrillation and other diseases, but they do not present any specific research data on this topic.

 3, There are significant amounts of irrelevant content in both the introduction and discussion parts that do not relate to the focus of the manuscript.

 4, The patient data in this study is all sourced from 2021, which is over a year old, and the sample size is small. This may affect the generalizability of the study's findings and the validity of the conclusions drawn from the data. 

5, Please define POAF when it occurred first.

 6, Page 2 line 76: The title here does not match the title of the manuscript.

 7, Line 151: Pleae replace " Graph 1. " with "Figure 1".

Author Response

Dear reviewer,

Thank you for your valuable comments. We applied all your suggestion to the manuscript.

On behalf of the authors.

Adriano Friganovic

R

Reviewer 2 Report (Previous Reviewer 1)

The authors addressed my comments

Author Response

Dear reviewer,

Thank you for your approval. We are happy that we succeeded to apply all your suggestion to the manuscript.

On behalf of the authors.

Adriano Friganovic

Round 2

Reviewer 1 Report (Previous Reviewer 2)

Dear Authors, Thank you for the quick reply. Your response to all suggestions is grateful. However, I would like to make one more suggestion, Page 4 line 134-142: can the authors organize this data into a table format?

Author Response

Dear reviewer,

We are very happy that we succeeded to follow your valuable suggestions. We applied your additional comment to the manuscript.

On behalf of the authors.

Adriano Friganovic

This manuscript is a resubmission of an earlier submission. The following is a list of the peer review reports and author responses from that submission.

Round 1

Reviewer 1 Report

- The study lacks novelty. It does not add new data. 

- The introduction is too long and not focused

- The methods are not clear: PAOF has to be defined.  it is not clear  when the participants answered the questionnaire.  What did this  questionnaire include? 

- The discussion section is not focused. Many un relevant data were mentioned

- The paper needs to be re written: introduction should be shortened and focused. Methods should be detailed: defining POAF, describing the questionnaire and how it was integrated in the study. Results section should indicate which data was  collected from the questionnaire or from the medical reports. Discussion section should focus on the mean findings and should describe how the results of the questionnaire  can help to improve nursing practice and quality of care for cardiac surgery patients with regard to daily activities for patients (as stated in conclusions)

Reviewer 2 Report

Presented data in this paper Prevalence of Postoperative Atrial Fibrillation and Nurses Practice – A Cross Sectional Study” show that atrial fibrillation was higher in the participants who had valve surgery compared to other cardiac surgeries. There was also an increase in atrial fibrillation in older participants. It is important for atrial fibrillation focused groups. However, several issues should be addressed before publication:

Comment 1: The title of the study is not appropriate as it does not address any impact of nursing practice on postoperative atrial fibrillation.   Comment 2: Abstract: Page 1 line 20: “We demonstrated a correlation between the prevalence of atrial fibrillation and other diseases.” The study does not provide evidence of a correlation between atrial fibrillation prevalence and other diseases as stated in the manuscript.   Comment 3: Materials and Methods: Page 3 lines 104 and 107: It is noted that the ethics committee approved the study in May, but data collection began in April, and it is suggested that the study should be double-checked for ethical compliance.   Comment 4: Results: Page 3 lines 141 and 142: “The mean age of the participants was 70-79 years”, The mean age of the participants should be expressed numerically and with the corresponding standard deviation or confidence interval, such as 75 ± 5 years.   Comment 5: Results: Page 5 lines 163: “The results of that association are shown in Table 3 and 4 and graphs 1 and 2.” the study states that the results of the correlation are presented in Table 3 and 4 and in Graphs 1 and 2, however, I did not find the relevant information in these tables and figures.
Comment 6: Results: Page 6 lines 176: Please change “Graph” to “Figure”. In a research paper, it is common to use "figure" to refer to charts or diagrams.